behaviour, developmental biology, evolution

avian eggshells, climate, life history, nest, temperature seasonality, water vapour conductance

**Author for correspondence:**
Marie R. G. Attard
e-mail: marie.attard@rhul.ac.uk

# Climate variability and parent nesting strategies influence gas exchange across avian eggshells

Marie R. G. Attard[1,2] and Steven J. Portugal[1,3]

[1]Department of Biological Sciences, School of Life and Environmental Sciences, Royal Holloway University of London, Egham, Surrey TW20 0EX, UK
[2]School of Engineering and Innovation, Open University, Milton Keynes MK7 6AA, UK
[3]The Natural History Museum, Tring, Herts HP23 6AP, UK

MRGA, 0000-0002-8509-3677; SJP, 0000-0002-2438-2352

Embryo survival in birds depends on a controlled transfer of water vapour and respiratory gases through the eggshell, and this exchange is critically sensitive to the surrounding physical environment. As birds breed in most terrestrial habitats worldwide, we proposed that variation in eggshell conductance has evolved to optimize embryonic development under different breeding conditions. This is the first study to take a broad-scale macro-ecological view of avian eggshell conductance, encompassing all key avian taxonomic groups, to assess how life history and climate influence the evolution of this trait. Using whole eggs spanning a wide phylogenetic diversity of birds, we determine that body mass, temperature seasonality and whether both parents attend the nest are the main determinants of eggshell conductance. Birds breeding at high latitudes, where seasonal temperature fluctuations are greatest, will benefit from lower eggshell conductance to combat temporary periods of suspended embryo growth and prevent dehydration during prolonged incubation. The nest microclimate is more consistent in species where parents take turns incubating their clutch, resulting in lower eggshell conductance. This study highlights the remarkable functional qualities of eggshells and their importance for embryo survival in extreme climates.

## 1. Introduction

Adaptive diversification across species typically occurs amidst an array of distinct ecological niches and environments and is a key driver in the development of novel functional traits to enhance the fitness of an organism [1]. The evolution of a new trait may provide the adaptive potential to exploit a resource that was not previously possible, or interact with its environment in a new way without a specific change in the external environment [2]. Close association between certain traits and a species environment and life history can therefore point to probable causes of trait divergence [3]. Traits can evolve rapidly over several generations or slowly over millions of years in accordance with environmental rates of change [4]. Species persistence during abrupt climate change will, therefore, depend on their ability to rapidly respond and adapt to novel environmental conditions [5]. Individual species will either move to more favourable conditions, tolerate or adapt to their changed environment, or go extinct [6]. Understanding the evolutionary history and diversification of functional traits closely linked to reproductive success will help predict how species will react to these new environmental pressures.

Foremost, the survival of any species is reliant on having a viable embryo. One crucial step in understanding avian responses to environmental differences over evolutionary time is a better appreciation of factors shaping avian incubation and their subsequent influence on the embryo [7]. Birds have evolved multiple functional traits to improve offspring survival in the nest: arguably

one of the most important is the eggshell. Most bird species lose 10–20% of their fresh egg mass over the incubation period through the passive diffusion of water vapour through the eggshell to the ambient air [8]. Eggs that lose too much water during incubation frequently do not hatch due to desiccation [9], while embryos that do not lose enough water from the egg experience respiratory problems or drown [10]. Maintaining a controlled loss of water from inside the egg to the external environment while allowing sufficient exchange of respiratory gases is therefore essential for normal embryo development and hatching.

Birds are highly diversified and widely distributed, occupying every continent on Earth and every terrestrial habitat within it [11]. Some birds breed in extremely inhospitable environments, such as cold and dry regions [12], deserts [13], moist wetlands [14] and high altitudes [15]. Among these are ground-nesting birds in alpine or Arctic/Antarctic regions that must cope with unpredictable wind, precipitation and snow conditions, with ambient temperatures fluctuating from below freezing to over 45°C [16]. Avian embryos in such cold regions will freeze to death if left unattended by their parents [17]. Desert birds that breed in the Sahara, Arabian and Kalahari regions face extreme physiological challenges to conserve water and avoid dehydration for the eggs, adults and hatchlings [18]. In contrast with dry, xeric environments, eggs exposed to high precipitation are prone to rain-induced suffocation [19]. The major challenges for birds breeding in high-altitude regions like the Himalayas is the low barometric pressure and high solar radiation, which can result in desiccation of egg contents and overheating of the embryo [15]. Species living at such environmental extremes must adapt behaviourally or physiologically at each stage of their breeding cycle if they are to produce viable offspring [20].

Water vapour conductance through the avian eggshell, herein referred to as conductance or $G_{H_2O}$, is influenced by the properties of the eggshell (e.g. pore length, functional pore area and eggshell cuticle) and humidity and gas composition of the surrounding environment. Species that incubate their eggs buried [21], in dry [22] or wet environments [23], or at high altitudes [15] have particularly unusual vapour pressure gradients, yet are still able to maintain water loss within acceptable limits. $G_{H_2O}$ may be optimized to suit particular environments through changes in nest-site preferences, eggshell structures and incubation behaviours [14], making eggs and their species-specific conductance ideal model systems for understanding how trait selection varies over time during diversification.

Predicting $G_{H_2O}$ of a species is not straight forward, as multiple ecological factors must be taken into account. For example, brood-chambers of burrow-nesting birds are often permanently saturated with water vapour, resulting in a low water vapour pressure (favouring enhanced conductance) and longer incubation periods (favouring reduced conductance) [24]. Inter-species differences in $G_{H_2O}$ thus can only be untangled by considering the contribution of multiple life-history traits and the phylogenetic history of the lineage. A study across 141 non-passerine species detected differences in $G_{H_2O}$ between nest types and parental incubation behaviours [25], emphasizing the importance of maintaining a suitable nest microclimate for optimum egg-water loss. However, it is unknown whether a similar relationship between conductance and nesting behaviour is expected in the passerines, which comprise over 6000 species and represent almost 60% of all living birds [26]. Moreover, previous studies have typically focused on either (i) one group of birds (e.g. gulls), with the goal to look for micro-adaptations between closely related species [27], or (ii) eggs of 'extreme nesters' such as desert-nesting Bedouin fowl (*Gallus domesticus*) [28] and grey gulls (*Larus modestus*) [29], water-nesting grebes and divers [30] and marsh-nesting black terns (*Chlidonias niger*) [31]. The role of life-history and environmental factors in the evolution of avian eggshell conductance thus requires a large-scale comparative analysis encompassing all key taxonomic groups.

Our aim was to evaluate how climate and life history influence $G_{H_2O}$ across a wide taxonomic distribution of birds spanning 28 avian orders, after accounting for the effects of adult body mass and phylogeny. Previous comparative analyses of eggshell conductance have not corrected for allometric effects of body mass [25], which can hide potentially important adaptive information relating to the environment and nesting behaviour of the species. Based on previous findings, we predicted $G_{H_2O}$ would be primarily explained by body mass. By contrast, we predict that mass-independent conductance ($RG_{H_2O}$) would be primarily associated with traits known to affect nest humidity, including climate, nest location and type.

## 2. Material and methods

### (a) Egg samples and preparation

In total, 365 bird species were included in this study. Conductance of whole emptied eggs at the Natural History Museum, Tring (NHM, UK) was established using the standard protocol of measuring the decrease in egg mass as a result of water loss over consecutive days, in eggs kept in constant moisture-free conditions [32]. $G_{H_2O}$ measured using whole eggshells is preferable to eggshell fragments as shell thickness and porosity varies between different regions of an egg [33]; therefore, we only used values from whole eggs in this study.

Eggs were prepared by gently cleaning the surface, filling the egg with water then sealing the blow hole (see electronic supplementary material). Eggs were placed in an acrylic desiccator cabinet (ThermoFisher Scientific, Nalgene, catalogue number: 5317-0070) inside a constant temperature thermocabinet (Porkka, Hertfordshire, UK) at $30 \pm 1$°C. Temperature was monitored via a logtag analyser every 10 min (Loggershop, Bournemouth, Dorset, UK). Self-indicating silica gel (Merck, Honenbrunn, Germany, catalogue number: 101 969) were placed in the desiccator to remove all moisture. Any loss in egg mass was entirely due to the diffusion of water vapour via the shell pores [34]. The first 24 h can give unexpectedly high mass loss values as the outer shell surface dries out [35]. Therefore, the eggs were left 24 h before being weighed to the nearest 0.1 mg (Sartorius, Göttingen, Germany), then were returned to the desiccator. Eggs were weighed at the same time of day on 3 successive days to give two values of 24 h mass loss ($M_{H_2O}$). Species $G_{H_2O}$ was then calculated, as described in the electronic supplementary material.

Species mean $G_{H_2O}$ values of whole eggs reported in the literature ($n = 188$) were incorporated if specimens had been measured under constant conditions (temperature and humidity) and followed protocols used in the present study. $G_{H_2O}$ measures from whole fresh eggs (unemptied or water-filled) and museum (water-filled) eggs were combined as $G_{H_2O}$ does not differ significantly between these treatments [36]. Mean $G_{H_2O}$ values reported in the literature were corrected to standard barometric pressure (1 ATM) at 30°C (see electronic supplementary material).

## (b) Life-history and ecological data

We collated data on 18 key life-history traits that have previously been hypothesized to play a role in the evolution of avian conductance in addition to climate variables (table 1). These data were extracted from multiple sources detailed in the FigShare repository (doi:10.6084/m9.figshare.12490559). Major sources are detailed in section (e) of electronic supplementary material. Only 13 predictors were included in the analysis due to collinearity (see electronic supplementary material). The phylogenetic generalized least-squares (PGLS) method was used to test the evolutionary association between whole eggshell $G_{H_2O}$ life-history traits, within a phylogenetic context [37]. In this procedure, closely related species are assumed to have more similar traits because of their shared ancestry and consequently will produce more similar residuals from the least-squares regression line. By taking into account the expected covariance structure of these residuals, modified slope and intercept estimates are generated that account for interspecific autocorrelation due to phylogeny.

Prior to updated avian phylogenies based on genomic DNA, near-passerines was a term given to tree-dwelling birds (within the conventional non-passerines) that were traditionally believed to be related to Passeriformes due to ecological similarities. In this study Pterocliformes (sandgrouse), Columbiformes (pigeons), Cuculiformes (cuckoos), Caprimulgiformes (nightjars) and Apodiformes (swifts, hummingbirds) were defined as near-passerines. All passerines and near-passerines are land birds and have altricial and nidicolous (stay within the nest) chicks, while non-passerine chicks vary in their mode of development and include water and land birds [38]. Sandgrouse are an exception as they have precocial young and are not tree-dwelling [39]. In respect to nest architecture, most passerines build open-cup nests, though some build more elaborate dome structures with roofs [40]. Dome nests, however, are more common among passerines than non-passerines and are particularly frequent among very small passerines [41]. Although these groups are no longer recognized as near-passerines, this definition was used here to distinguish between ecologically profound differences among birds.

Avian phylogenetic trees were constructed online (http://www.birdtree.org) from the complete avian phylogeny of Jetz et al. [42] and used the primary backbone tree of Hackett et al. [43]. Ten thousand trees were constructed and statistical analyses were performed in the program R, v. 3.6.1 (R Software, Vienna, Austria, http://www.R-project.org). All quantitative variables (except absolute median latitude, annual temperature and temperature range) were $log_{10}$-transformed prior to phylogenetic analysis to reduce skewness [44].

As body mass affects all aspects of animal biology and ecology [45], our initial set of phylogenetic analysis account for adult body mass by including this variable as a predictor of $log(G_{H_2O})$. We repeated our phylogenetic analysis using mass-corrected $G_{H_2O}$ as the response variable, herein called relative $G_{H_2O}$ ($RG_{H_2O}$), thereby removing adult body mass as a predictor. $RG_{H_2O}$ values were computed as residuals from a PGLS regression of $log(G_{H_2O})$ on log(body mass) (slope = 0.53 ± 0.03 s.e.; intercept = −0.69 ± 0.12 s.e.; $\lambda = 0.68$; electronic supplementary material, figure S1). Using this second series of models, we can ask how well one or more life-history traits results in higher or lower $G_{H_2O}$ than is expected for a given body mass of the adult bird.

Phylogenetic signal in $G_{H_2O}$ and $RG_{H_2O}$ was measured by Pagel's lambda ($\lambda$) [46] using the phylosig function in the package 'phytools' [47] to determine to what extent related species were more likely to share similar conductance values than species drawn randomly from a tree. The phylosig function was used to test the hypothesis that Pagel's $\lambda$ is different from 0. To test the alternative hypothesis (that Pagel's $\lambda$ is less than 1), we computed the difference in the log-likelihood ratio of the lambda model (phylosig function) and Brownian motion model (brownie.lite

function), then compared it to a chi-squared ($\chi^2$) distribution with 1 degree of freedom. PGLS models were fitted using the phylolm function in the package 'phylolm' [48]. We ran the full model containing all traits as predictor variables, then used the pdredge function from the package MuMIn [49] to fit all possible model combinations with a maximum of five predictors following protocols by Powney et al. [50], in addition to a null model comprising only the intercept. The best subset of models had an AICc (Akaike's information criterion adjusted for low sample size) within two of the model with the lowest AICc [51]. Conditional model averaging was then used to identify parameter estimates and importance for each trait present in at least one of the subset models [52].

## 3. Results

In total, we used over 2533 eggs from 364 species to assess diversification in conductance across the avian phylogeny. These species span across 85 families and represent 28 of the 49 extant avian orders. Overall, bird species in Australia, North America and South America had higher $log(G_{H_2O})$ and $RG_{H_2O}$ than species in Africa, Europe and Asia (electronic supplementary material, figure S2). $G_{H_2O}$ was highest for large flightless birds (e.g. ostriches (Struthio camelus) (106.99 mg day$^{-1}$ Torr$^{-1}$)), nightjars (Caprimulgiformes $0.55 ± 0.19$ mg day$^{-1}$ Torr$^{-1}$) and songbirds (Passeriformes $0.74 ± 0.05$ mg day$^{-1}$ Torr$^{-1}$). $G_{H_2O}$ was also high for aquatic birds (e.g. common loons (Gavia immer) 98.82 mg day$^{-1}$ Torr$^{-1}$), kiwis (Southern brown kiwi (Apteryx australis) 26.22 mg day$^{-1}$ Torr$^{-1}$) and penguins (Sphenisciformes $22.66 ± 5.45$ mg day$^{-1}$ Torr$^{-1}$). Viewing total phylogenetic variation in this trait (figures 1a and 2a) revealed that $log(G_{H_2O})$ and $RG_{H_2O}$ were typically lower in passerines and near-passerines, than non-passerines (figure 1b).

### (a) Phylogenetic correlation

Phylogenetic signal for $log(G_{H_2O})$ and $RG_{H_2O}$ (table 2) was significantly different from 0 (i.e. no phylogenetic signal) ($p < 0.001$) and 1 (i.e. the Brownian explanation) ($p < 0.001$), meaning that while there is an effect of phylogeny on conductance, it is influenced by evolutionary processes that are weaker than would be seen with a Brownian motion model of trait evolution. Phylogenetic signal was high for $Log(G_{H_2O})$ ($\lambda = 0.96$), showing that closely related species exhibit similar eggshell conductance prior to accounting for differences in body mass, and this biological similarity decreases as the evolutionary distance between species increases. Phylogenetic signal was intermediate for $RG_{H_2O}$ ($\lambda = 0.55$), suggesting that phylogeny and other selective pressures (e.g. those associated with species life history or climate) are important in determining eggshell conductance, after accounting for differences in species body mass.

### (b) Life history and climate influence conductance across birds

Adult body mass and temperature seasonality were the strongest predictors of $log(G_{H_2O})$ across all birds based on conditionally averaged models (electronic supplementary material, table S1). $Log(G_{H_2O})$ was significantly higher among heavier species ($z = 18.40$, $p < 0.001$; figures 1c and 3a) since initial egg mass increases with adult body mass ($n = 251$, $r^2 = 0.89$,

**Table 1.** Putative predictions and definitions for 13 possible explanations for variation in water vapour conductance ($G_{H_2O}$) in birds.

| predictor | hypothesis | definition |
|---|---|---|
| body mass | as adult body mass is correlated to egg mass, heavier birds will have higher $G_{H_2O}$ due to greater egg surface area | mean body mass (g) of adult birds |
| clutch size | evaporation from multiple eggs will create a nest atmosphere of greater humidity and reduced water vapour transfer, so $G_{H_2O}$ should be higher for species with larger clutches | number of eggs per brood, measured as geometric mean of the typical minimum and maximum clutch size |
| calcium content | eggshells of calcium-poor species are expected to be thinner, less dense and more porous than calcium-rich species, and thus facilitate higher $G_{H_2O}$ | (1) calcium-rich: species that ingest mollusc shells, fish, shellfish, calcareous grit, calcareous ash or bones<br>(2) calcium-poor: species with primarily insectivorous or granivorous diet |
| egg maculation | maculated eggs are expected to have lower $G_{H_2O}$ than immaculate eggs to reduce the risk of desiccation | (1) immaculate: no spotting or markings on eggshell surface<br>(2) maculation: maculation present on eggshell surface |
| nest type | fully enclosed nests have less air movement than semi-enclosed and exposed nests, facilitating greater $G_{H_2O}$ | (1) exposed: nest is open above and has no side walls (no nest, scrape, saucer, platform, heap)<br>(2) semi-enclosed: nest is partially open and has side walls (cup, bowl, pendant, sphere, dome, pouch)<br>(3) enclosed: nest is entirely enclosed (cavity, burrow, crevice) |
| nest location | nests above ground have lower risk of flooding or water accumulation, therefore will have lower $G_{H_2O}$ | (1) ground: nest location in or on the ground, or floating on water<br>(2) tree: nest located in tree, bush, shrub, wall, cave roof, or attached to reed<br>(3) cliff: nest located on cliff |
| nest lining | incorporation of nest lining will better insulate the egg, therefore will have higher $G_{H_2O}$ | (1) lined: nest lining is always or sometimes present<br>(2) not lined: nest lining is absent |
| habitat | among open nesting species, more direct sunlight reaches eggs in open habitats and experience greater air movement around the nest than closed habitats; open-nesting species in open habitats will have lower $G_{H_2O}$ than in closed habitats | (1) open: breeds in desert, grassland, open water, open moorland, low shrubs, rocky habitats, seashores and cities<br>(2) semi-open: breeds in open shrubland and bushland, scattered bushes, parkland, forest edge<br>(3) dense: breeds in forest with a closed canopy, or in the lower vegetation strata of dense thickets, shrubland, mangroves or marshland |
| incubating parent | nest vapour pressure will decrease when the parent leaves the nest uncovered, which is more likely to occur if incubation is not shared between parents, resulting in lower $G_{H_2O}$ | (1) shared: contact incubation of eggs by two adults<br>(2) not shared: contact incubation of eggs by single adult |
| mode of development | higher $G_{H_2O}$ may contribute to improving the use of nutritional support by the embryo of precocial species by removing excess water, thus resulting in increased development at hatching | (1) altricial: newly born young are relatively immobile, naked, and usually require care and feeding by the parents<br>(2) precocial: newly born young are relatively mobile, covered in feathers, and independent |
| parental care | the eggs of species that provide biparental care are expected to have higher $G_{H_2O}$ as nest humidity and temperature can be better maintained when both parents assist | (1) uniparental: the brood is provisioned and/or defended by one adult<br>(2) biparental: the brood is provisioned and/or defended by two adults |

(Continued.)

**Table 1.** (*Continued.*)

| predictor | hypothesis | definition |
| --- | --- | --- |
| parental contact | the wet incubating parent returning to the nest will increase the nest's humidity, thus are excepted to have higher $G_{H_2O}$ | (1) wet plumage: adults returned habitually to the nest with wet plumage; this included species that feed on freshwater or marine prey or use nests built on water |
| | | (2) dry plumage: adults did not return habitually to the nest with wet plumage |
| temperature seasonality | eggs incubated in environments with highly variable temperature will experience lower $G_{H_2O}$ as high temperature seasonality occurs in cooler environments | average temperature seasonality (BIO4) of breeding/resident range, based on WorldClim v1 data |
| precipitation seasonality | eggs incubated in environments with highly variable precipitation will experience higher $G_{H_2O}$ to combat temporary periods of excessive rain | average precipitation seasonality (BIO15) of breeding/resident range, based on WorldClim v1 data |

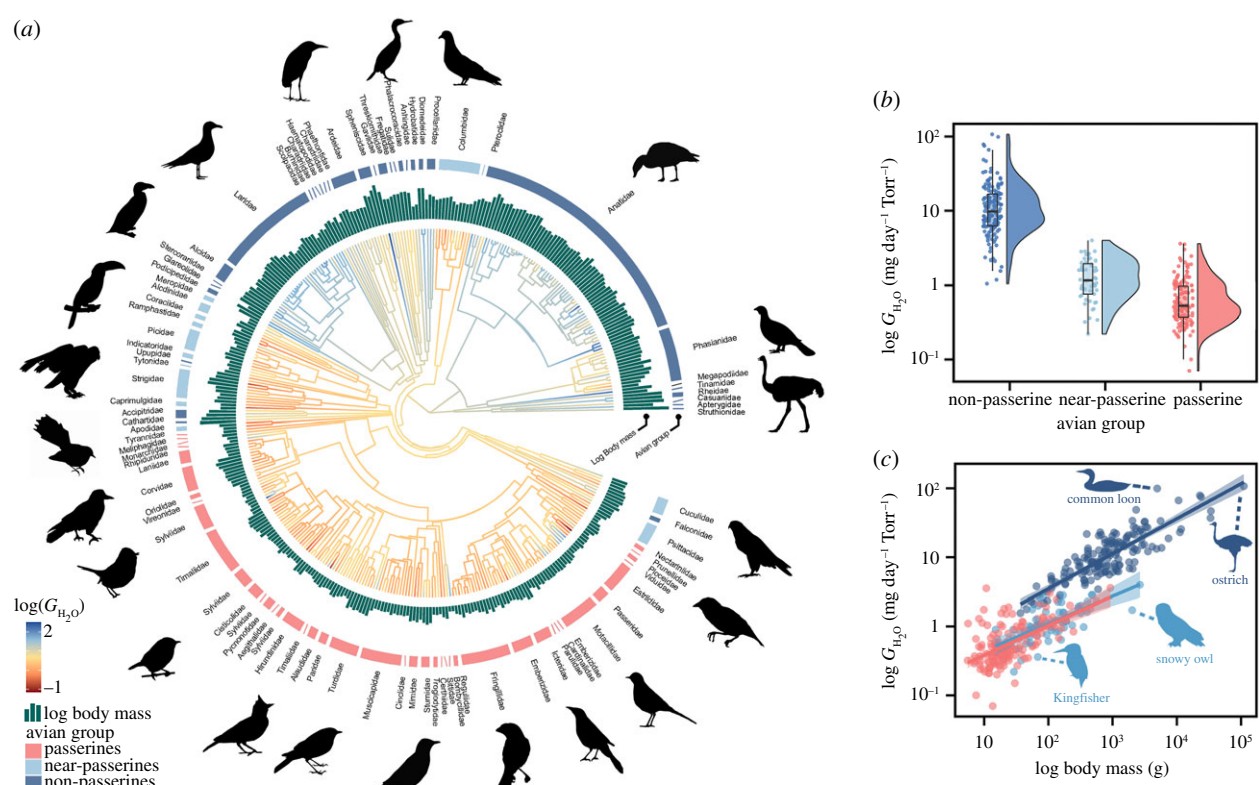

**Figure 1.** Relationship between conductance of whole eggs and ecological variables for 364 bird species. (*a*) Phylogenetic tree from which water vapour conductance ($G_{H_2O}$) data were obtained. The bar plot around the phylogeny represents the only significant predictors of log($G_{H_2O}$) in conditionally averaged models. Conditional model averaging was used to obtain a single average model when more than one PGLS model was best ranked (i.e. more than one model with $\Delta$AICc < 2 from the top-ranked model). Branch colours show the diversification in log($G_{H_2O}$) across the phylogeny and ancestral trait estimates. $G_{H_2O}$ is plotted as a function of (*b*) avian group and (*c*) adult body mass (g) within each of the three avian groups. Silhouette illustrations came from PhyloPic (http://phylopic.org), contributed by various authors under public domain licence. (Online version in colour.)

$p < 0.001$ [52]). Log($G_{H_2O}$) was negatively associated with increased temperature seasonality across all birds ($z = 2.13$, $p = 0.03$; figures 2*b* and 3*b*). Temperature seasonality is defined here as the amount of temperature variation over a given year (or averaged years) based on the standard deviation of monthly temperature averages [53]. There was also a weaker yet significant effect of dietary calcium, nest location, mode of development, shared incubation and parental contact among top-ranked models (electronic supplementary material, table S2).

Temperature seasonality ($z = 2.20$, $p = 0.03$) and whether contact incubation was shared among parents ($z = 2.22$, $p = 0.03$) were significant in conditionally averaged models after accounting for adult body mass ($RG_{H_2O}$) (electronic supplementary material, table S3; figure 2). $RG_{H_2O}$ overall decreased with temperature seasonality (figures 2*b* and 3*b*).

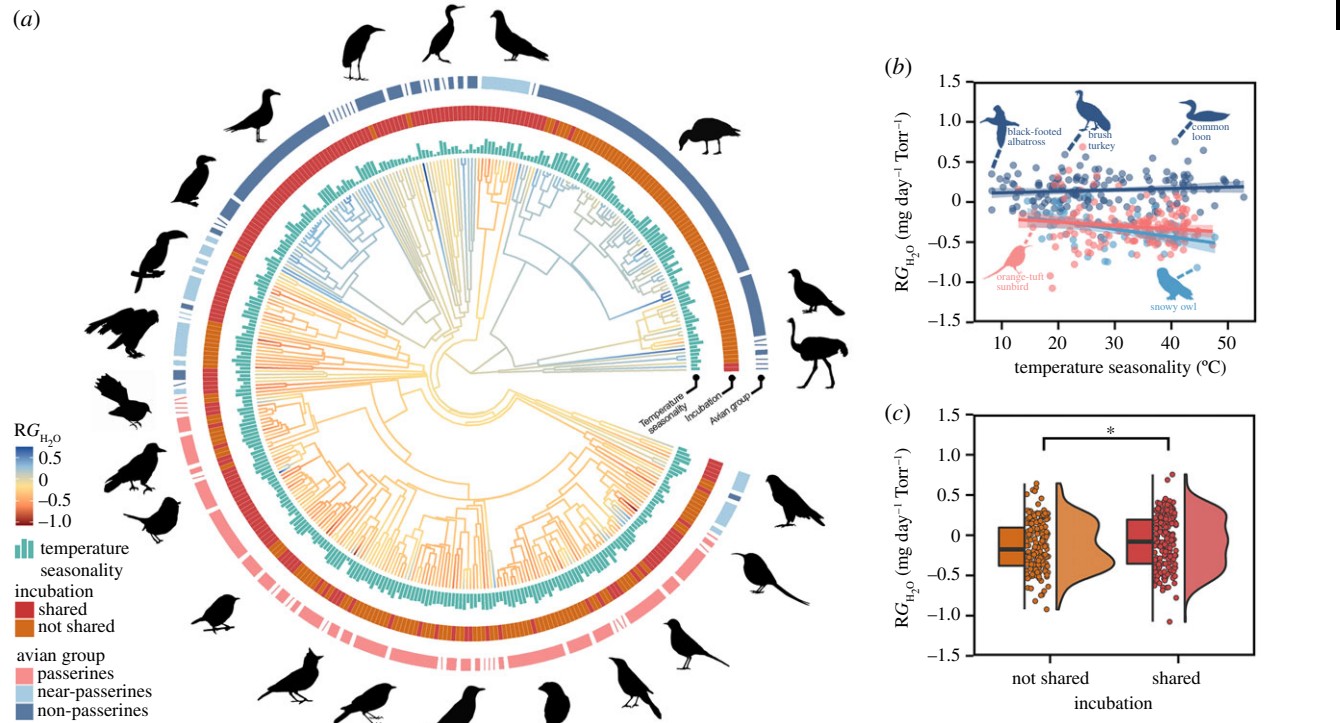

**Figure 2.** Relationship between conductance of whole eggs and ecological variables for 364 bird species. (a) Phylogenetic tree of residual water vapour conductance ($RG_{H_2O}$). Bar plots and rings around the phylogeny represent significant predictors of $RG_{H_2O}$ in conditionally averaged models. Conditional model averaging was used to obtain a single average model when more than one PGLS model was best ranked (i.e. more than one model with $\Delta AICc < 2$ from the top-ranked model). Branch colours show the diversification in $RG_{H_2O}$ across the phylogeny and ancestral trait estimates. $RG_{H_2O}$ is plotted as a function of (b) temperature seasonality within each avian group and (c) whether both parents incubate the eggs. In the hybrid box plot, species $RG_{H_2O}$ are shown as filled circles, vertical lines indicate the median, box shows the interquartile range (IQR) and the whiskers are $1.5 \times$ IQR (distribution is shown as histograms). p-values are given in asterisks, with *less than 0.05, **less than 0.01 and ***less than 0.001. Silhouette illustrations came from PhyloPic (http://phylopic.org), contributed by various authors under public domain license. (Online version in colour.)

**Table 2.** Estimates of phylogenetic signal in eggshell water vapour conductance ($G_{H_2O}$) in all birds. Phylogenetic signal was analysed separately for $\log_{10}$-transformed $G_{H_2O}$ ($\log(G_{H_2O})$) and residual water vapour conductance ($RG_{H_2O}$). The p-value tests the null hypothesis of no phylogenetic signal ($\lambda = 0$) and Brownian motion model ($\lambda = 1$) of evolution.

| response variable | Pagel's $\lambda$ | log-likelihood | log-likelihood for $\lambda = 0$ | log-likelihood for $\lambda = 1$ | p for $\lambda = 0$ | p for $\lambda = 1$ |
|---|---|---|---|---|---|---|
| $\log(G_{H_2O})$ | 0.96 | −74.39 | 590.76 | −125.64 | <0.001 | <0.001 |
| $RG_{H_2O}$ | 0.55 | 27.20 | 258.50 | −92.27 | <0.001 | <0.001 |

$RG_{H_2O}$ was higher in species where both parents incubate the clutch (figure 2c). Dietary calcium, mode of development, nest location and parental contact showed weaker but significant correlations with $RG_{H_2O}$ among top-ranked models (electronic supplementary material, table S4). $RG_{H_2O}$ was higher in species with calcium-rich diets, precocial young, parents that return to the nest with wet plumage and ground nesters compared to tree nesters (electronic supplementary material, figure S6). Based on conditionally averaged models for $G_{H_2O}$ and $RG_{H_2O}$, eggshell conductance across birds is primarily influenced by adult body mass, temperature seasonality and parent incubation strategies.

## 4. Discussion

This study focused on one performance trait—conductance—of modern avian eggshells to better understand how birds have achieved high ecological diversity. We identified the importance of phylogeny, physiology (body mass and mode of development), behaviour (diet, parental incubation strategies and nest location) and climate in the evolution of this trait. This study is the first to identify a broad-scale reduction in eggshell conductance where temperature seasonality increases. Regions with greater temperature seasonality experience a greater range in temperatures over the course of a year and correlate with an organisms' temperature tolerance breadth [54]. Increased temperature seasonality occurs further from the equator and is associated with a decline in annual temperature, precipitation and day length [55]. A comparative study on 139 bird species found that adults inhabiting low and seasonally variable temperatures had lower basal metabolic rate after removing the effects of body mass [56]. In the light of this, it appears possible that eggshells are already preparing the embryo for adulthood, with respect to their environment and breeding biology.

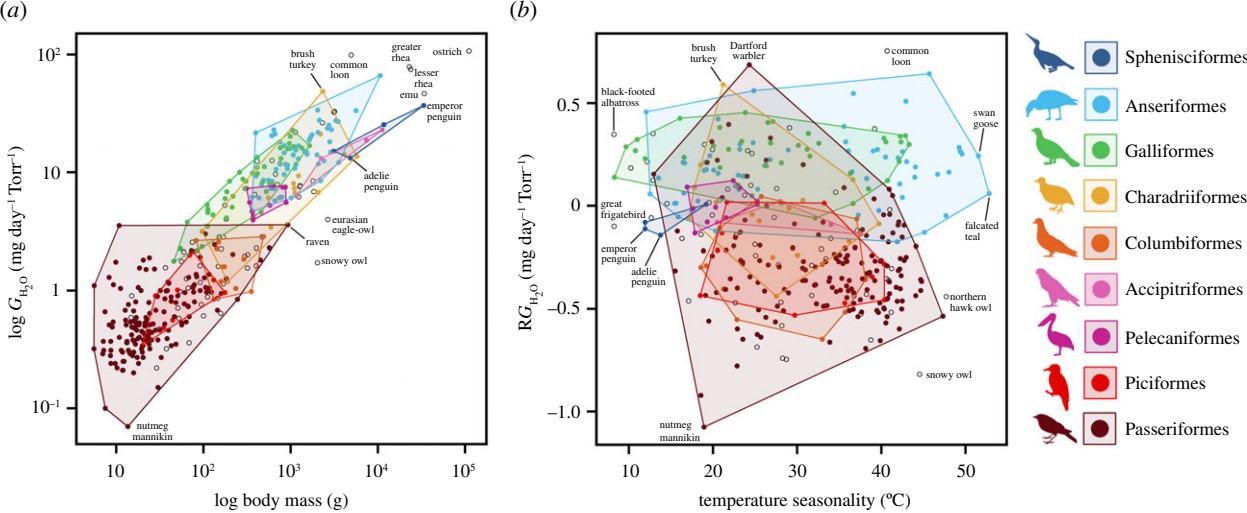

**Figure 3.** Partitioning of variation of water vapour conductance ($G_{H_2O}$) among avian orders. Dots (one per species) ($n = 364$) show the distribution of $\log(G_{H_2O})$ as a function of (*a*) adult body mass (g) and (*b*) residual water vapour conductance ($RG_{H_2O}$) as a function of temperature seasonality. The minimum convex hull is plotted for all species within a subset of avian orders. Silhouette illustrations came from PhyloPic (http://phylopic.org), contributed by various authors under public domain license. (Online version in colour.)

Amniotic embryos adjust their metabolic activity and active cell division in response to varying environmental conditions, and by doing so, alter their period of development [57]. Reproductive strategies to prolong the egg state are most diverse in reptiles and less varied in birds and mammals that provide more parental care [58]. Even so, the low metabolic rate expected for embryos incubated in highly seasonal climates would favour a reduction in conductance to prolong their incubation period.

Broad-scale geographical trends in $RG_{H_2O}$ identified here may be the result of long-term evolutionary responses or short-term physiological modifications [59]. Evolutionary adaptation would involve changes in $G_{H_2O}$ over (rather than within) generations when natural selection acts on genetic variants while acclimatization would involve reversible changes to $G_{H_2O}$ that can happen gradually (greater than 1 day) in response to the recent environment [60]. Intraspecific variation in $G_{H_2O}$ has been reported across altitude [61,62] and humidity [63,64] gradients of multiple species, but the timeframe in which $G_{H_2O}$ diversification has taken place is unknown. Some studies propose that rapid evolution of eggshell structure from exposure to novel environments is unlikely [65,66] and is instead compensated by behavioural modifications of the parents. Other studies find that incubation behaviour does not significantly modulate conductance [67], so adaptive responses must be accomplished by changes in eggshell structure [63].

Birds are seemingly capable of short-term and instantaneous physiological adjustments in shell structure in response to environmental variation. Pigeons (*Columba livia*) bred for several years within an environmentally controlled room experienced approximately 30% lower $G_{H_2O}$ than predicted when exposed to high temperature and low humidity over a short period [68]. Similarly, domestic chickens (*Gallus domesticus*) bred at high elevation for multiple generations produced eggshells with a 30% higher $G_{H_2O}$ within two months of being translocated to low altitudes [69]. In other species, $G_{H_2O}$ did not change when individuals were transferred to higher altitudes [70] or were exposed to natural seasonal changes in humidity [71], suggesting there is variation in the plasticity of a species response. Identifying the speed of the response in eggshell parameters to novel

environments across multiple species will be very informative in determining climate change effects on bird species and their breeding.

We found that conductance across birds was also dependent on nest location, whether parents alternate nest attendance, and whether the parent returns to the nest with wet plumage, corroborating previous studies [25,72]. Shared incubation between two parents allows one of them to be relieved from incubation to feed while the other incubates the egg, thus allowing the eggs to be covered at all times [73]. Clutches that are incubated by both parents encounter less variation in egg temperature than clutches that are incubated by a single parent [74] and thus, are expected to have higher eggshell conductance. Water added to the nest by parents can be many orders of magnitude higher than water lost by the eggs [75]. Consequently, $RG_{H_2O}$ is significantly higher in species where parents return the nest with wet plumage [25]. Eggs laid on the ground, in a burrow, mound or on floating vegetation are subject to higher humidity than arboreal nesters, leading to eggshell adaptations that promote water loss. Common loons (*Gavia immer*), for example, had the highest $RG_{H_2O}$ of the species investigated. This may be attributed to their high eggshell porosity [76] in response to building nests on or near the water where transpiration of water is high, and nest materials can be wet [64]. Nest location and whether parents return to the nest with wet or dry plumage was significant in most top models where these predictors were included, but this effect was weak compared to life-history traits retained in conditionally averaged models. Combined, our results demonstrate that different behavioural strategies used by parents to alter nest humidity have contributed to the evolution of conductance among birds.

Variation in the incubation period across the altricial-precocial spectrum reflects a trade-off between embryo growth rate and degree of maturity when hatched. Precocial species take up to two times longer to incubate an egg of the same size as altricial species, but are far more developed when they hatch [77]. For eggs of the same mass, precocial species incur a higher total energy cost than altricial species because the embryo is larger for a longer period during incubation [78]. Consequently, eggs of species with fast

(precocial) growing offspring had significantly higher $RG_{H_2O}$ than those of species with slow (altricial) growing offspring based on top-ranked models. As higher conductance enables greater gas exchange, this may optimize embryo access to high energy content in precocial eggs [79], thus resulting in a more developed chick at birth. $RG_{H_2O}$ in passerines was found here to be particularly low, likely because they have altricial young, whereas non-passerines consist of precocial and altricial species.

Data accessibility. Data are publicly available in the FigShare repository, including specimen and species-specific water vapour conductance, life histories and sources used in this study (doi:10.6084/m9.fig share.12490559) [80]. Tables for all PGLS analyses and sources for figure illustrations are available in the electronic supplementary material.

Authors' contributions. M.R.G.A.: conceptualization, data curation, formal analysis, investigation, methodology, project administration, validation, visualization, writing-original draft, writing-review and editing; S.J.P.: conceptualization, formal analysis, funding acquisition, investigation, methodology, project administration, resources, supervision, writing-review and editing

All authors gave final approval for publication and agreed to be held accountable for the work performed therein.

Competing interests. We declare we have no competing interests.

Funding. This project and M.R.G.A. was funded by a Research Project (grant no. RPG-2018-332) from the Leverhulme Trust, awarded to S.J.P.

Acknowledgements. We are grateful to Douglas Russell at The Natural History Museum Tring for his generous assistance in working with the eggshell collection, and for useful discussions. We thank Craig White for useful discussions and providing code for phylogenetic comparative analysis, and Stephanie McClelland, Jennifer Cantlay and Jack Thirkell for their comments on early drafts.

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
