## [Peer Review File · Proceedings of the Royal Society B: Biological Sciences]

Review History

RSPB-2020-2685.R0 (Original submission)

Review form: Reviewer 1

Recommendation

Accept with minor revision (please list in comments)

Scientific importance: Is the manuscript an original and important contribution to its field?

Excellent

General interest: Is the paper of sufficient general interest?

Good

Quality of the paper: Is the overall quality of the paper suitable?

Excellent

Is the length of the paper justified?

Yes

Should the paper be seen by a specialist statistical reviewer?

No

Do you have any concerns about statistical analyses in this paper? If so, please specify them explicitly in your report.

No

It is a condition of publication that authors make their supporting data, code and materials available - either as supplementary material or hosted in an external repository. Please rate, if applicable, the supporting data on the following criteria.

Is it accessible?

No

Is it clear?

N/A

Is it adequate?

N/A

Do you have any ethical concerns with this paper?

No

Comments to the Author

This manuscript, by Attard & Portugal, considers the inter-specific variation in eggshell conductance across a global sample of 364 species, finding that conductance (i.e. propensity for water loss during incubation) relates to adult body mass, temperature seasonality, and mode of development. This manuscript is very clearly written and examines an aspect of avian development not often considered at this scale.

My comments are all minor – most are superficial, with a few requests for increased clarity about the methods. I think this manuscript is very, very close to publication-ready: well done to the authors!

As a general comment, though, I would be curious about the authors' views of any sampling biases underlying this dataset, given that they are making macroevolutionary conclusions on a comparatively small (albeit phylogenetically diverse) sample of species. Tring's collections are heavily biased towards former British colonies, and any Western museum collection will have better temperate sampling than tropical; the availability of eggs in the destructive collections would surely have put further taxonomic and biogeographic constraints on this study. This is of course unavoidable, but I'd be curious to see the authors' speculations about how a more phylogenetically representative sample might change their results.

I'd also be curious about the intraspecific variation in GH2O, given that the authors seem to have a large enough sample to investigate this. My understanding is that PGLS doesn't allow measurement error (though MCMCgIimm does, if the authors are so inclined to test this), but even a partition of the variance in the database between intra- and inter-specific effects would reassure the reader both that this is a sensible variable to investigate at the comparative level and that the biases in comparative methods inherent in traits with high measurement error would not apply here. (See e.g. Silvestro et al. 2015 ME&E, <https://besjournals.onlinelibrary.wiley.com/doi/10.1111/2041-210X.12337>, or Ives et al. 2007 Systematic Biology <https://academic.oup.com/sysbio/article/56/2/252/1687174>)

Anyway, enough speculation. Minor comments:

L31: Consider "terrestrial habitats".

L70-73: This seems like an overstatement. Something along the lines of "one crucial step in understanding avian responses to environmental differences over evolution time is a better appreciation of..." would be less inflammatory but convey the same point?

L84: Again, every *terrestrial* habitat.

L86: Do you mean birds that are both alpine and Arctic/Antarctic or do you mean either alpine or Arctic/Antarctic? The syntax is unclear.

L168: Where exactly did the life-history data come from? If it's at all a reasonable number of sources, the original compilers of the data would probably appreciate the citations; even if listing the major sources is untenable, more information is needed. (Even just stating that the major sources are detailed in section e of the supplement would be useful to the reader!)

L196-198: I understand why some would find an analysis of residuals to be intuitively easier to understand than the multivariate regression described a few sentences previously, but a regression of residuals can be statistically flawed in many circumstances. See for example <https://besjournals.onlinelibrary.wiley.com/doi/full/10.1046/j.1365-2656.2002.00618.x> (Freckleton 2002 JAE). It sounds like you ran both analyses, which is okay, but tread carefully. In particular, I had trouble sometimes distinguishing which results were based on the multiple regression and which were based on the regression-of-residuals. (Moreover, if the results of these two analyses aren't similar, you have a problem with underlying correlations of your predictor variables, and would need to err on the side of your multiple regression.)

L233: The interpretation of $\lambda < 1$ is tricky (see for example <https://www.carlboettiger.info/2013/10/11/is-it-time-to-retire-pagels-lambda.html> for a clear explanation of some of the issues of this metric). $\lambda = 1$ indicates that Brownian motion is a perfect fit for the distribution of the data. $\lambda < 1$ could indicate a weak effect of phylogeny under Brownian motion, or it could indicate a *strong* effect of phylogeny under another macroevolutionary mode; it's impossible to tell from the analysis conducted here.

L236: Is strongly constrained by phylogeny *under the assumption of Brownian motion*. (I'm not bickering with the analysis here - what you did is perfectly standard. The phrasing of the interpretation just needs to be more precise.)

Figure 1: I assume these are distribution maps for breeding ranges only?

Figure 2: This is a very attractive figure (as is Figure 3). How did you define near-passerine?

Figure 2: Are the Creative Commons licenses all CC0 1.0? If so this should be more clearly stated; if not, the details of the licenses should be listed somewhere (in the supplement), as per the legal conditions of using phylopic.

Supplement, line 93: What do you mean by "extrapolated" here?

L94-S: Does the FigShare link (which didn't work for me, presumably by design) list all sources of this information? (Same question for the "monographs" in L100-S and the other sources mentioned in this paragraph.)

L135-S: Missing quotation mark (or possibly an extra quotation mark, depending on what you were intending)

L161-163-S: I don't understand what you're trying to convey here. Was your releveling of factors somehow dependent on the results you obtained?

Table S2: Consider making the font size on this table smaller (or decreasing the number of columns); numbers and words spanning multiple lines make this very difficult to read. (Same can be said about Table S4, though to a lesser extent.)

Review form: Reviewer 2

Recommendation

Accept with minor revision (please list in comments)

Scientific importance: Is the manuscript an original and important contribution to its field?

Acceptable

General interest: Is the paper of sufficient general interest?

Good

Quality of the paper: Is the overall quality of the paper suitable?

Excellent

Is the length of the paper justified?

No

Should the paper be seen by a specialist statistical reviewer?

Yes

Do you have any concerns about statistical analyses in this paper? If so, please specify them explicitly in your report.

Yes

It is a condition of publication that authors make their supporting data, code and materials available - either as supplementary material or hosted in an external repository. Please rate, if applicable, the supporting data on the following criteria.

Is it accessible?

Yes

Is it clear?

Yes

Is it adequate?

Yes

Do you have any ethical concerns with this paper?

No

Comments to the Author

The present study provides evidence that eggshell functionality evolves in relation to environmental pressures. The manuscript is very clear and it makes for a very interesting read. I only have some minor suggestions that I would like to see addressed before it is published.

- It would be useful to know why (or why not) would one expect that in passerines the relationship between conductance and nest type, parental incubation be different to that previously reported by Portugal et al. 2014. In other words, is there anything fundamentally different between passerine egg or nest architecture/ incubation behaviour and to non-passerine reproductive traits?

- It was not obvious until reading the results that the study spanned half of the avian orders and not only passerines (which I first thought, therefore the above comment). It would be useful to explain this since the introduction or methods.

- line130: change "to effect" with "to affect" (affect is the verb)

- How many species were included in the measurements of conductance? (mention in the section "egg samples and preparation")

- Although the sampling effort performed is commendable, sampled species account for ~4% of all bird species, this will always be a problem with comparative studies. How did the authors deal with the issue that results from having a large amount of missing data in the phylogeny, i.e. the outcome of the analyses is very sensitive to the inclusion of just a few extra species?

I found the discussion slightly lengthy, perhaps some sections can be condensed. For example, last paragraph of page 13 (and first of page 14), also paragraph about calcium sources on page 14.

Figure 1.- I am a little skeptical about the usefulness of information presented here, perhaps is my personal unfamiliarity with this type of mapping, how was this performed? How reliable is it? Moreover, the continental differences are not further discussed or contrasted with the finding that temperature seasonality had an effect on conductance. However, seasonality would vary with latitude but not merely differ between North America and Europe I imagine? What can one take as a message from this information?

Figure 2. This figure is impressive and aesthetically appealing but it is overloaded with information and difficult to interpret. The tree for example, displays the ancestral state estimation plus 5 variables. The magnitude of the bars from the two first variables is hard to perceive. Could this figure be split?

Decision letter (RSPB-2020-2685.R0)

27-Nov-2020

Dear Dr Attard:

I am writing to inform you that your manuscript RSPB-2020-2685 entitled "Climate variability and mode of development influence gas exchange across avian eggshells" has, in its current form, been rejected for publication in Proceedings B.

This action has been taken on the advice of referees, who have recommended that substantial revisions are necessary. With this in mind we would be happy to consider a resubmission, provided the comments of the referees are fully addressed. However please note that this is not a provisional acceptance.

Sincerely,
 Dr Sasha Dall
 mailto: proceedingsb@royalsociety.org

Associate Editor

Comments to Author:

Thank you for submitting your manuscript "Climate variability and mode of development influence gas exchange across avian eggshells" to Proceedings B. I have now received two reviews and evaluated the manuscript myself. While we all find the topic interesting, a number of issues have been raised that need to be addressed. Please pay particular attention to the questions regarding sampling biases made by Reviewer 1 and comments by reviewer 2 about highlighting the novelty of your current work and differentiating it from your previous research.

Reviewer(s)' Comments to Author:

Referee: 1

Comments to the Author(s)

This manuscript, by Attard & Portugal, considers the inter-specific variation in eggshell conductance across a global sample of 364 species, finding that conductance (i.e. propensity for water loss during incubation) relates to adult body mass, temperature seasonality, and mode of development. This manuscript is very clearly written and examines an aspect of avian development not often considered at this scale.

My comments are all minor – most are superficial, with a few requests for increased clarity about the methods. I think this manuscript is very, very close to publication-ready: well done to the authors!

As a general comment, though, I would be curious about the authors' views of any sampling biases underlying this dataset, given that they are making macroevolutionary conclusions on a comparatively small (albeit phylogenetically diverse) sample of species. Tring's collections are heavily biased towards former British colonies, and any Western museum collection will have better temperate sampling than tropical; the availability of eggs in the destructive collections would surely have put further taxonomic and biogeographic constraints on this study. This is of course unavoidable, but I'd be curious to see the authors' speculations about how a more phylogenetically representative sample might change their results.

I'd also be curious about the intraspecific variation in GH₂O, given that the authors seem to have a large enough sample to investigate this. My understanding is that PGLS doesn't allow measurement error (though MCMCglmm does, if the authors are so inclined to test this), but even a partition of the variance in the database between intra- and inter-specific effects would reassure the reader both that this is a sensible variable to investigate at the comparative level and that the biases in comparative methods inherent in traits with high measurement error would not apply here. (See e.g. Silvestro et al. 2015 ME&E, <https://besjournals.onlinelibrary.wiley.com/doi/10.1111/2041-210X.12337>, or Ives et al. 2007 Systematic Biology <https://academic.oup.com/sysbio/article/56/2/252/1687174>)

Anyway, enough speculation. Minor comments:

L31: Consider "terrestrial habitats".

L70-73: This seems like an overstatement. Something along the lines of "one crucial step in understanding avian responses to environmental differences over evolution time is a better appreciation of..." would be less inflammatory but convey the same point?

L84: Again, every *terrestrial* habitat.

L86: Do you mean birds that are both alpine and Arctic/Antarctic or do you mean either alpine or Arctic/Antarctic? The syntax is unclear.

L168: Where exactly did the life-history data come from? If it's at all a reasonable number of sources, the original compilers of the data would probably appreciate the citations; even if listing the major sources is untenable, more information is needed. (Even just stating that the major sources are detailed in section e of the supplement would be useful to the reader!)

L196-198: I understand why some would find an analysis of residuals to be intuitively easier to understand than the multivariate regression described a few sentences previously, but a regression of residuals can be statistically flawed in many circumstances. See for example <https://besjournals.onlinelibrary.wiley.com/doi/full/10.1046/j.1365-2656.2002.00618.x> (Freckleton 2002 JAE). It sounds like you ran both analyses, which is okay, but tread carefully. In particular, I had trouble sometimes distinguishing which results were based on the multiple regression and which were based on the regression-of-residuals. (Moreover, if the results of these two analyses aren't similar, you have a problem with underlying correlations of your predictor variables, and would need to err on the side of your multiple regression.)

L233: The interpretation of $\lambda < 1$ is tricky (see for example <https://www.carlboettiger.info/2013/10/11/is-it-time-to-retire-pagels-lambda.html> for a clear explanation of some of the issues of this metric). $\lambda = 1$ indicates that Brownian motion is a perfect fit for the distribution of the data. $\lambda < 1$ could indicate a weak effect of phylogeny under Brownian motion, or it could indicate a *strong* effect of phylogeny under another macroevolutionary mode; it's impossible to tell from the analysis conducted here.

L236: Is strongly constrained by phylogeny *under the assumption of Brownian motion*. (I'm not bickering with the analysis here - what you did is perfectly standard. The phrasing of the interpretation just needs to be more precise.)

Figure 1: I assume these are distribution maps for breeding ranges only?

Figure 2: This is a very attractive figure (as is Figure 3). How did you define near-passerine?

Figure 2: Are the Creative Commons licenses all CC0 1.0? If so this should be more clearly stated; if not, the details of the licenses should be listed somewhere (in the supplement), as per the legal conditions of using phylopic.

Supplement, line 93: What do you mean by "extrapolated" here?

L94-S: Does the FigShare link (which didn't work for me, presumably by design) list all sources of this information? (Same question for the "monographs" in L100-S and the other sources mentioned in this paragraph.)

L135-S: Missing quotation mark (or possibly an extra quotation mark, depending on what you were intending)

L161-163-S: I don't understand what you're trying to convey here. Was your releveling of factors somehow dependent on the results you obtained?

Table S2: Consider making the font size on this table smaller (or decreasing the number of columns); numbers and words spanning multiple lines make this very difficult to read. (Same can be said about Table S4, though to a lesser extent.)

Referee: 2

Comments to the Author(s)

The present study provides evidence that eggshell functionality evolves in relation to environmental pressures. The manuscript is very clear and it makes for a very interesting read. I only have some minor suggestions that I would like to see addressed before it is published.

- It would be useful to know why (or why not) would one expect that in passerines the relationship between conductance and nest type, parental incubation be different to that previously reported by Portugal et al. 2014. In other words, is there anything fundamentally different between passerine egg or nest architecture/ incubation behaviour and to non-passerine reproductive traits?

- It was not obvious until reading the results that the study spanned half of the avian orders and not only passerines (which I first thought, therefore the above comment). It would be useful to explain this since the introduction or methods.

- line130: change “to effect” with “to affect” (affect is the verb)
- How many species were included in the measurements of conductance? (mention in the section “egg samples and preparation”)
- Although the sampling effort performed is commendable, sampled species account for ~4% of all bird species, this will always be a problem with comparative studies. How did the authors deal with the issue that results from having a large amount of missing data in the phylogeny, i.e. the outcome of the analyses is very sensitive to the inclusion of just a few extra species?

I found the discussion slightly lengthy, perhaps some sections can be condensed. For example, last paragraph of page 13 (and first of page 14), also paragraph about calcium sources on page 14.

Figure 1.- I am a little skeptical about the usefulness of information presented here, perhaps is my personal unfamiliarity with this type of mapping, how was this performed? How reliable is it? Moreover, the continental differences are not further discussed or contrasted with the finding that temperature seasonality had an effect on conductance. However, seasonality would vary with latitude but not merely differ between North America and Europe I imagine? What can one take as a message from this information?

Figure 2. This figure is impressive and aesthetically appealing but it is overloaded with information and difficult to interpret. The tree for example, displays the ancestral state estimation plus 5 variables. The magnitude of the bars from the two first variables is hard to perceive. Could this figure be split?

Author's Response to Decision Letter for (RSPB-2020-2685.R0)

See Appendix A.

RSPB-2021-0823.R0

Review form: Reviewer 2 (Liliana D'Alba)

Recommendation

Accept as is

Scientific importance: Is the manuscript an original and important contribution to its field?

Excellent

General interest: Is the paper of sufficient general interest?

Acceptable

Quality of the paper: Is the overall quality of the paper suitable?

Excellent

Is the length of the paper justified?

Yes

Should the paper be seen by a specialist statistical reviewer?

No

Do you have any concerns about statistical analyses in this paper? If so, please specify them explicitly in your report.

No

It is a condition of publication that authors make their supporting data, code and materials available - either as supplementary material or hosted in an external repository. Please rate, if applicable, the supporting data on the following criteria.

Is it accessible?

Yes

Is it clear?

Yes

Is it adequate?

Yes

Do you have any ethical concerns with this paper?

No

Comments to the Author

It is my opinion that after revision this paper is ready for publication. The authors did a superb job addressing all my previous comments and replying to each question in great detail. Thus, I learned something new during the reviewing process.

Decision letter (RSPB-2021-0823.R0)

07-May-2021

Dear Dr Attard

I am pleased to inform you that your Review manuscript RSPB-2021-0823 entitled "Climate variability and parent nesting strategies influence gas exchange across avian eggshells" has been accepted for publication in Proceedings B.

The referee(s) do not recommend any further changes. Therefore, please proof-read your manuscript carefully and upload your final files for publication. Because the schedule for publication is very tight, it is a condition of publication that you submit the revised version of your manuscript within 7 days. If you do not think you will be able to meet this date please let me know immediately.

To upload your manuscript, log into <http://mc.manuscriptcentral.com/prsb> and enter your Author Centre, where you will find your manuscript title listed under "Manuscripts with Decisions." Under "Actions," click on "Create a Revision." Your manuscript number has been appended to denote a revision.

You will be unable to make your revisions on the originally submitted version of the manuscript. Instead, upload a new version through your Author Centre.

1) A text file of the manuscript (doc, txt, rtf or tex), including the references, tables (including captions) and figure captions. Please remove any tracked changes from the text before submission. PDF files are not an accepted format for the "Main Document".

2) A separate electronic file of each figure (tiff, EPS or print-quality PDF preferred). The format should be produced directly from original creation package, or original software format. Please note that PowerPoint files are not accepted.

3) Electronic supplementary material: this should be contained in a separate file from the main text and the file name should contain the author's name and journal name, e.g. `authorname_procb_ESM_figures.pdf`

All supplementary materials accompanying an accepted article will be treated as in their final form. They will be published alongside the paper on the journal website and posted on the online figshare repository. Files on figshare will be made available approximately one week before the accompanying article so that the supplementary material can be attributed a unique DOI. Please see: <https://royalsociety.org/journals/authors/author-guidelines/>

4) Data-Sharing and data citation

It is a condition of publication that data supporting your paper are made available. Data should be made available either in the electronic supplementary material or through an appropriate repository. Details of how to access data should be included in your paper. Please see <https://royalsociety.org/journals/ethics-policies/data-sharing-mining/> for more details.

<http://datadryad.org/submit?journalID=RSPB&manu=RSPB-2021-0823> which will take you to your unique entry in the Dryad repository.

Once again, thank you for submitting your manuscript to Proceedings B and I look forward to receiving your final version. If you have any questions at all, please do not hesitate to get in touch.

Sincerely,

Dr Sasha Dall

Associate Editor

Board Member

Comments to Author:

Thank you for submitting your manuscript "Climate variability and mode of development influence gas exchange across avian eggshells" to Proceedings B. I have now received feedback from a reviewer concerning the revisions of your manuscript and evaluated it myself. The reviewers and I agree that you have successfully addressed all concerns.

Reviewer(s)' Comments to Author:

Referee: 2

Comments to the Author(s).

It is my opinion that after revision this paper is ready for publication. The authors did a superb job addressing all my previous comments and replying to each question in great detail. Thus, I learned something new during the reviewing process.

Decision letter (RSPB-2021-0823.R1)

14-May-2021

Dear Dr Attard

I am pleased to inform you that your manuscript entitled "Climate variability and parent nesting strategies influence gas exchange across avian eggshells" has been accepted for publication in Proceedings B.

If you are likely to be away from e-mail contact please let us know. Due to rapid publication and an extremely tight schedule, if comments are not received, we may publish the paper as it stands. If you have any queries regarding the production of your final article or the publication date please contact procb_proofs@royalsociety.org

Data Accessibility section

Open Access

Paper charges

Sincerely,

Proceedings B

Appendix A

Response to Referees

We thank the editor and both reviewers for their valuable suggestions and insight on our manuscript. The suggestions offered by the reviewers have been immensely helpful, and we also appreciate your insightful comments to better communicate the novelty of our research.

We have addressed all reviewer comments and have thoroughly revised the manuscript, incorporating most suggested changes or explaining why we have adopted a different strategy. Additionally, we have corrected a slight oversight in our R script and have set a more conservative cut-off for correlation coefficients between predictors for the phylogenetic comparative analysis. This has slightly changed the results, however, the overall message and findings of the paper remains the same. All specimen and species eggshell conductance data, conductance and life-history sources and R scripts used in this paper can now be accessed by the reviewers using the private link (<https://figshare.com/s/ab01f5a20d00673018b3>) on Figshare. We have added the Figshare DOI to the paper, which will become activate upon publication.

Comments from the editors and reviewers are in italics and are responses are in bold. Text that has been updated in the main document as part of our responses is highlighted in yellow. We also refer to line numbers from the revised manuscript in our responses. Amendments to the main text based on changes in our results are highlighted in green.

Associate Editor

Comments to Author:

Thank you for submitting your manuscript “Climate variability and mode of development influence gas exchange across avian eggshells” to Proceedings B. I have now received two reviews and evaluated the manuscript myself. While we all find the topic interesting, a number of issues have been raised that need to be addressed. Please pay particular attention to the questions regarding:

We’re pleased yourself and the reviewers found our paper interesting, and we believe we have fully addressed all the issues that have been raised.

1) *Sampling biases made by Reviewer 1*

This has now been addressed in the manuscript. Our response can be found below and also in response to Reviewer 1, question 1 under “General comments”.

The reviewer makes an excellent point, that there is potential for bias. This potential for bias in the destructive collection at the Natural History Museum was the primary driver for encompassing data from the literature. The 188

species from the literature have a worldwide distribution, along with adding phylogenetic diversity to the study also.

2) *Comments by Reviewer 2 about highlighting the novelty of your current work and differentiating it from your previous research.*

The novelty of this research is two-fold: it presents the first large-scale phylogenetic comparative assessment of eggshell conductance in birds (encompassing all key taxonomic groups) and its association with different life-history traits. Secondly, our phylogenetic comparative analysis uses regression-of-residuals to account for differences in adult body mass, allowing us to address the adaptive importance of specific traits of a species after allometric effects are considered.

Previous comparative research on eggshell conductance has focused on non-passerines, ignoring 60% of bird species, given passerines make a substantial component of the avian kingdom. These prior studies on non-passerines did not correct for differences in body mass of the adults, which potentially hides critical adaptive information relating to the environment and nesting behaviour of the species (see L111-122). Moreover, previous studies have typically focused either on (i) one group of birds, e.g., gulls, with the goal to look for micro-adaptations between closely related species, or (ii) extreme examples of nest environments such as grebes and marsh-nesting black terns (*Chlidonias niger*). Thus, this is the first study to take a broad-scale macro-ecological holistic view of avian eggshell conductance and look for commonalities and patterns across a broad-scale of bird species, all within a phylogenetic framework.

As such, we have now updated the Abstract and Introduction (see below) to emphasise the need for large-scale comparative analysis encompassing all key avian taxonomic groups to determine how life-history and climate influence the evolution of eggshell conductance.

Abstract:

“This is the first study to take a broad-scale macro-ecological view of avian eggshell conductance, encompassing all key avian taxonomic groups, to assess how life history and climate influence the evolution of this trait.” (L33-36).

Introduction:

“A study across 141 non-passerine species detected differences in G_{H_2O} between nest types and parental incubation behaviours [25], emphasising the importance of maintaining a suitable nest microclimate for optimum egg-water loss. However, it is unknown whether a similar relationship between conductance and nesting behaviour is expected in the passerines, which comprise over 6,000 species and represent almost 60% of all living birds [26]. Moreover, previous studies have typically focused either on (i) one group of birds, e.g., gulls, with the goal to look for micro-adaptations between closely related species [27], or (ii) eggs of ‘extreme nesters’ such as desert-nesting Bedouin fowl (*Gallus domesticus*) [28] and grey gulls (*Larus modestus*) [29],

water-nesting grebes and divers [30] and marsh-nesting black terns (*Chlidonias niger*) [31]. The role of life-history and environmental factors in the evolution of avian eggshell conductance thus requires a large-scale comparative analysis encompassing all key taxonomic groups.” (L117-129).

“Previous comparative analyses of eggshell conductance have not corrected for allometric effects of body mass [25], which can hide potentially important adaptive information relating to the environment and nesting behaviour of the species.” (L133-136)

Reviewer 1

Comments to the Author(s)

This manuscript, by Attard & Portugal, considers the inter-specific variation in eggshell conductance across a global sample of 364 species, finding that conductance (i.e. propensity for water loss during incubation) relates to adult body mass, temperature seasonality, and mode of development. This manuscript is very clearly written and examines an aspect of avian development not often considered at this scale.

My comments are all minor – most are superficial, with a few requests for increased clarity about the methods. I think this manuscript is very, very close to publication-ready: well done to the authors!

Thank you very much for your kind words and positive feedback. It’s much appreciated. We’re glad you found the work clearly written, and believe it to be close to publication-ready.

General comments:

1) As a general comment, though, I would be curious about the authors’ views of any sampling biases underlying this dataset, given that they are making macroevolutionary conclusions on a comparatively small (albeit phylogenetically diverse) sample of species. Tring’s collections are heavily biased towards former British colonies, and any Western museum collection will have better temperate sampling than tropical; the availability of eggs in the destructive collections would surely have put further taxonomic and biogeographic constraints on this study. This is of course unavoidable, but I’d be curious to see the authors’ speculations about how a more phylogenetically representative sample might change their results.

The reviewer makes an excellent point, that there is potential for bias. This potential for bias in the destructive collection at the Natural History Museum was the primary driver for encompassing data from the literature. The 188 species from the literature have a worldwide distribution, along with adding phylogenetic diversity to the study also. Thus, eggshell conductance measures included in our study were based on specimens at the Natural History Museum that we measured in our laboratory (176 species) as well as mean species conductance values from the literature (188 species). As such, over half of the species in our dataset are derived from multiple sources

worldwide, which has broadened the spatial distribution of species incorporated in this study.

We agree that there are some biogeographical constraints by incorporating a large portion of British species. It is worth noting, however, that many British species have substantial geographical distributions (barn owls, peregrine falcons, house sparrows, ospreys etc), even though the eggs themselves (in the present study) for those species were gathered from the United Kingdom.

2) I'd also be curious about the intraspecific variation in G_{H_2O} , given that the authors seem to have a large enough sample to investigate this. My understanding is that PGLS doesn't allow measurement error (though MCMCglmm does, if the authors are so inclined to test this), but even a partition of the variance in the database between intra- and inter-specific effects would reassure the reader both that this is a sensible variable to investigate at the comparative level and that the biases in comparative methods inherent in traits with high measurement error would not apply here. (See e.g. Silvestro et al. 2015 ME&E, <https://besjournals.onlinelibrary.wiley.com/doi/10.1111/2041-210X.12337>, or Ives et al. 2007 Systematic Biology <https://academic.oup.com/sysbio/article/56/2/252/1687174>)

We thank Reviewer 1 for their suggestion to run a MCMCglmm to assess variation in the dataset between intra- and inter-specific effects. We agree that this would greatly benefit the paper, and clarify the contribution that intraspecific variation is making to our overall findings. Unfortunately, we do not have specimen-specific values for any species obtained from the literature, as only mean species values were presented within the paper and thus available to us. As such, we are unable to run such an analysis on the full data set. We reviewed the papers from the literature, and many were from decades ago with authors who are no longer with us. Due to this, we decided it wouldn't be fruitful to try and track down these full data sets from contacting the authors.

Of course, however, we do have all values in full for our species that we directly measured in the laboratory. We have applied MCMCglmm to 176 species from our dataset, where we measured conductance of eggs directly. When accounting for intra-specific variation in eggshell conductance, body mass was the only significant predictor of $\log(G_{H_2O})$ in passerines based on MCMCglmm. This concurs with our findings based on PGLS on 364 species.

As the MCMCglmm results do not cover a broad taxonomic range due to the limited number of non-passerines with specimen-specific values ($n=1$), using these samples alone does not meet the main objective of our study. Therefore, we decided not to include MCMCglmm results in the revised manuscript.

Table 1. The effects of life-history traits on eggshell conductance in 176 bird species using MCMCglimm. Shown are posterior estimates of the effect size in the full model (plus the 95% credibility intervals, and the probability that the effect is different from the null hypothesis).

Predictor	Posterior mean	Lower 95% CI	Upper 95% CI	Effect sample	pMCMC
Intercept	-1.19	-1.72	-0.71	1000.00	<0.001
Body mass	0.52	0.44	0.60	1000.00	<0.001
Calcium ^a	-0.01	-0.07	0.06	1000.00	0.85
Clutch size	0.08	-0.17	0.34	704.60	0.53
Egg maculation ^b	-0.01	-0.08	0.06	1000.00	0.92
Nest type					
-Semi-enclosed ^c	-0.05	-0.16	0.07	1000.00	0.41
-Enclosed ^c	-0.05	-0.16	0.07	768.60	0.44
-Enclosed ^d	-0.00	-0.08	0.07	1000.00	0.95
Nest lining ^e	-0.06	-0.13	0.02	758.40	0.13
Nest location					
-tree ^f	-0.06	-0.13	0.02	1000.00	0.13
-cliff ^g	-0.08	-0.20	0.04	1000.00	0.17
-cliff ^g	-0.02	-0.15	0.10	1000.00	0.71
Habitat					
-semi-open ^h	0.01	-0.05	0.08	1000.00	0.68
-dense ^h	0.03	-0.03	0.11	1000.00	0.37
-dense ⁱ	0.02	0.04	0.08	1000.00	0.54
Shared incubation ^j	-0.04	-0.10	0.02	1000.00	0.19
Development mode ^k	0.17	-0.16	0.45	864.90	0.26
Parental contact ^l	-0.05	-0.16	0.05	1000.00	0.34
Temperature					
seasonality	0.00	-0.01	0.00	1289.50	0.46
Precipitation					
seasonality	0.05	-0.11	0.19	1000.30	0.42

^aCalcium poor was the reference group.

^bImmaculate egg was the reference group.

^cExposed nest was the reference group.

^dSemi-enclosed nest was the reference group.

^eLined nest was the reference group.

^fGround nest was the reference group.

^gTree nest was the reference group.

^hopen habitat was the reference group.

ⁱSemi-open habitat was the reference group.

^jShared incubation was the reference group.

^kPrecocial was the reference group.

^lDry plumage was the reference group.

Minor comments:

1) L31: Consider “terrestrial habitats”.

Done. Reworded to “terrestrial habitats” (L31), as suggested.

2) L70-73: *This seems like an overstatement. Something along the lines of “one crucial step in understanding avian responses to environmental differences over evolution time is a better appreciation of...” would be less inflammatory but convey the same point?*

A very good point, and we have amended accordingly as follows:

“One crucial step in understanding avian responses to environmental differences over evolutionary time is a better appreciation of factors shaping avian incubation and their subsequent influence on the embryo [7].” (L70-73)

3) L84: *Again, every *terrestrial* habitat.*

Done. Reworded to “terrestrial habitat” (L84).

4) L86: *Do you mean birds that are both alpine and Arctic/Antarctic or do you mean either alpine or Arctic/Antarctic? The syntax is unclear.*

Thank you for spotting this, as we had been ambiguous. We have corrected the syntax to “Among these are ground nesting birds in alpine or Arctic/Antarctic regions” (L86-87).

5) L168: *Where exactly did the life-history data come from? If it’s at all a reasonable number of sources, the original compliers of the data would probably appreciate the citations; even if listing the major sources is untenable, more information is needed. (Even just stating that the major sources are detailed in section e of the supplement would be useful to the reader!)*

We have clarified this in the main manuscript and in the supplementary, to make sure it is absolutely clear where the life-history information came from. The life-history traits did come from multiple sources, but the main sources are stated in Supplementary Information only, due to the limited word count for the main manuscript.

As suggested by the reviewer, we have referred to section (e) of supplementary, in the main manuscript, so primary sources can be found. L180-182 now states “This data was extracted from multiple sources detailed in the Figshare repository (DOI: 10.6084/m9.figshare.12490559). Major sources are detailed in section (e) of the Supplementary Information.”

In section (e) of the Supplementary information (L101-108-S), we state that “Adult body mass, habitat, latitude and climate variables for all species were obtained from Sheard *et al.* [18]. Clutch size, fresh egg mass and incubation days were predominantly obtained from Myhrvold *et al.* [17] and eggshell

thickness (mm) was predominantly obtained from Schönwetter [19]. Eggshell maculation was categorised based on photos on several online databases (see dataset in Figshare repository). Other life-history traits were primarily gathered from *Handbook of Birds of the World Alive* [20], *Birds of the Western Palearctic* [21] and published scientific databases [17,22,23]. Any gaps in our dataset were then filled using monographs.”

The Figshare DOI will be activated once the paper is published. Reviewers can download all data and R scripts directly using the private Figshare link (<https://figshare.com/s/ab01f5a20d00673018b3>). Every source used to extract life-history data for each species can be found in the following spreadsheet:

- “species_eggshell_conductance_data_references.xlsx”

We have also uploaded this spreadsheet in our manuscript resubmission for the reviewers. Please refer to tab “Life-history ref list” and “Conductance literature ref list” in the spreadsheet for the full list of references for life-history and conductance values from the literature. The tab “species_data” contains all species-specific conductance and life-history data. The column “source_lifehistory” (column AV) cites all sources used to obtain life-history traits for each individual species.

6) L196-198: *I understand why some would find an analysis of residuals to be intuitively easier to understand than the multivariate regression described a few sentences previously, but a regression of residuals can be statistically flawed in many circumstances. See for example <https://besjournals.onlinelibrary.wiley.com/doi/full/10.1046/j.1365-2656.2002.00618.x> (Freckleton 2002 JAE). It sounds like you ran both analyses, which is okay, but tread carefully. In particular, I had trouble sometimes distinguishing which results were based on the multiple regression and which were based on the regression-of-residuals. (Moreover, if the results of these two analyses aren't similar, you have a problem with underlying correlations of your predictor variables, and would need to err on the side of your multiple regression.)*

Thank you for your suggestions. We ran both analyses, and found that the same predictors were present in the top-ranked models (AIC<2). Body mass was highly influential (Table S1 and S2) in predicting G_{H_2O} , so we ran PGLS on the regression-of-residuals to account for allometric effects, and to determine what contributed to the observed variation in eggshell conductance, after accounting for body mass.

To help distinguish between our results, the main predictors of $\log(G_{H_2O})$ and residual G_{H_2O} are now presented as two separate figures. Figure 1 now shows the most influential predictors for $\log(G_{H_2O})$, while figure 2 shows the most influential predictors for residual G_{H_2O} . Hopefully this is much clearer now, and avoids any ambiguity.

7) L233: *The interpretation of $\lambda < 1$ is tricky (see for example <https://www.carlboettiger.info/2013/10/11/is-it-time-to-retire-pagels-lambda.html> for a clear explanation of some of the issues of this metric). $\lambda = 1$ indicates that Brownian motion is a perfect fit for the distribution of the data. λ*

*< 1 could indicate a weak effect of phylogeny under Brownian motion, or it could indicate a *strong* effect of phylogeny under another macroevolutionary mode; it's impossible to tell from the analysis conducted here.*

Thank you for raising this point. We agree with the reviewer that we cannot tell from the analysis whether the effect is “strong” or “weak”. Following the reviewer’s suggestion, we have reworded L264-271 to: “Phylogenetic signal was high for $\text{Log}(G_{H_2O})$ ($\lambda=0.96$), showing that closely related species exhibit similar eggshell conductance prior to accounting for differences in body mass, and this biological similarity decreases as the evolutionary distance between species increases. Phylogenetic signal was intermediate for $R_{G_{H_2O}}$ ($\lambda=0.55$), suggesting that phylogeny and other selective pressures (e.g., those associated with species life-history or climate) are important in determining eggshell conductance, after accounting for differences in species body mass.”

Additionally, we have added in the manuscript how we tested if Pagel’s lambda is significantly different from 0 and 1: “The *phylosig* function was used to test the hypothesis that Pagel’s λ is different from 0. To test the alternative hypothesis (that Pagel’s λ is less than 1), we computed the difference in the log-likelihood ratio of the lambda model (*phylosig* function) and Brownian motion model (*brownie.lite* function), then compared it to a chi-squared (χ^2) distribution with 1 degree of freedom.” (L229-233).

We used the R script from Dr Liam Revell (<http://blog.phytools.org/2012/11/testing-for-pagels-10.html>) to test both hypotheses.

*8) L236: Is strongly constrained by phylogeny *under the assumption of Brownian motion*. (I’m not bickering with the analysis here – what you did is perfectly standard. The phrasing of the interpretation just needs to be more precise.)*

We agree, and thank you for highlighting this. We have made changes in the manuscript to clarify our interpretation, pasted below:

“Phylogenetic signal was high for $\text{Log}(G_{H_2O})$ ($\lambda=0.96$), showing that closely related species exhibit similar eggshell conductance prior to accounting for differences in body mass, and this biological similarity decreases as the evolutionary distance between species increases. Phylogenetic signal was intermediate for $R_{G_{H_2O}}$ ($\lambda=0.55$), suggesting that phylogeny and other selective pressures (e.g., those associated with species life-history or climate) are important in determining eggshell conductance, after accounting for differences in species body mass.” (L264-271)

9) Figure 1: I assume these are distribution maps for breeding ranges only?

The distribution maps were created using spatial data compiled by BirdLife International and Handbook of birds of the World. This encompasses the total range of each species, as unfortunately shapefiles of species breeding range alone are not available.

We have provided a description of how the global spatial distribution maps for eggshell conductance were created in section (i) of Supplementary Information and have upload the R script and all shapefiles used to create the eggshell conductance distribution map to Figshare. These files are compressed into the folder “spatial distribution traits”, and can be downloaded by reviewers using our private Figshare link (<https://figshare.com/s/ab01f5a20d00673018b3>). As the distribution map is not pivotal to our study, we have moved this figure to Supplementary information (Figure S2).

10) *Figure 2: This is a very attractive figure (as is Figure 3). How did you define near-passerine?*

We have added a description of near-passerines in L192-206 of the methods to be absolutely clear (see below). We do agree that these categories are somewhat contentious. However, we were still keen to explore these three groups separately due to possibility that eggshell conductance in near-passerines may more closely resemble passerines than non-passerines because of their ecological similarities.

“Prior to updated avian phylogenies based on genomic DNA, near-passerines was a term given to tree-dwelling birds (within the conventional non-passerines) that were traditionally believed to be related to Passeriformes due to ecological similarities. In this study Pteroclitiformes (sandgrouse), Columbiformes (pigeons), Cuculiformes (cuckoos), Caprimulgiformes (nightjars), and Apodiformes (swifts, hummingbirds) were defined as near-passerines. All passerines and near-passerines are land birds and have altricial and nidicolous (stay within the nest) chicks, while non-passerine chicks vary in their mode of development and include water and land birds [38]. Sandgrouse are an exception as they have precocial young and are not tree-dwelling [39]. In respect to nest architecture, most passerines build open-cup nests, though some build more elaborate dome structures with roofs [40]. Dome nests, however, are more common among passerines than non-passerines, and are particularly frequent among very small passerines [41]. Although these groups are no longer recognised as near-passerines, this definition was used here to distinguish between ecologically profound differences among birds.”

11) *Figure 2: Are the Creative Commons licenses all CC0 1.0? If so this should be more clearly stated; if not, the details of the licenses should be listed somewhere (in the supplement), as per the legal conditions of using phylopic.*

Yes, all images in Figure 1, 2 and 3 are provided under Creative Commons licenses (CC0 1.0 or CC by 3.0), or were created by the manuscript authors. In Supplementary Information, we have now provided a full list of sources for images from Figure 1, 2 and 3, and links to the original source (see subheading ‘(j) Silhouette sources’). We apologise for the oversight of leaving this important information out.

Supplementary Material

12) L93-S: What do you mean by “extrapolated” here?

These values were obtained from Supplementary material provided by Sheard *et al.* 2020. We have changed “extrapolated” to “obtained” in the text (L101-102-S) for clarification, and is pasted below.

“Adult body mass, habitat, latitude and climate variables for all species were obtained from Sheard *et al.* [18].”

13) L94-S: Does the Figshare link (which didn’t work for me, presumably by design) list all sources of this information? (Same question for the “monographs” in L100-S and the other sources mentioned in this paragraph.)

This is correct. By design, the Figshare DOI will not work until the paper is published. We are happy to share these data with the referees prior to publication so they can be assessed. The life-history spreadsheet (titled “species_eggshell_conductance_data_references.xlsx”) has been uploaded in the re-submission. This spreadsheet can also be downloaded by reviewers using our private Figshare link (<https://figshare.com/s/ab01f5a20d00673018b3>).

In the life-history spreadsheet, we list every source used in a separate column next to each species so readers can find the original source of information. The full reference list for life-history sources is also provided in a separate tab in the Excel spreadsheet. Most life-history data were obtained from sources stated in Supplementary information (under subheading ‘(e) Life-history and ecological data’, and see below), where available.

“Adult body mass, habitat, latitude and climate variables for all species were obtained from Sheard *et al.* [18]. Clutch size, fresh egg mass and incubation days were predominantly obtained from Myhrvold *et al.* [17] and eggshell thickness (mm) was predominantly obtained from Schönwetter [19]. Eggshell maculation was categorised based on photos on several online databases (see dataset in Figshare repository). Other life-history traits were primarily gathered from *Handbook of Birds of the World Alive* [20], *Birds of the Western Palearctic* [21] and published scientific databases [17,22,23]. Any gaps in our dataset were then filled using monographs.”

14) L135-S: Missing quotation mark (or possibly an extra quotation mark, depending on what you were intending)

We thank the reviewer for pointing this out. Extra quotation mark added.

15) L161-163-S: I don’t understand what you’re trying to convey here. Was your releveling of factors somehow dependent on the results you obtained?

We have now clarified this in the supplementary text. If there were more than 2 levels in a factor (e.g., the category “nest location” has three levels: ground, tree, cliff), the PGLS needs to be run again, whereby the levels of this factor are re-ordered so that the level specified by “ref” is first and the others are

moved down. In doing so, each level becomes the reference to be compared with the remaining levels in that factor. This allows pair-wise comparisons within each level to be assessed, as shown in the conditional averaged outputs. This does not influence the results of other factors included in the analysis. All R scripts used in this publication, including PGLS, will be available on Figshare. The private link to download R scripts prior to publication is <https://figshare.com/s/ab01f5a20d00673018b3>.

We have pasted our amendment to Supplementary Information below:

“When a multi-level predictor (i.e., a factor with more than two levels) was significant for conditional averaging, the “relevel” function was used to reorder the levels of that predictor and thereby change the reference variable for that predictor. The PGLS models were then rerun to obtain conditional averaging results for all multi-level predictor pair-wise comparisons. For example, the category “nest location” had three levels (ground, tree and cliff), thus the PGLS needs to be run again, whereby the levels of this factor are re-ordered so that the level specified as the reference is first, and the others are moved down. In doing so, each level becomes the reference to be compared with the remaining levels in that factor. This allows pair-wise comparisons within each level to be assessed, as shown in the conditional averaged outputs. This does not influence the results of other factors included in the analysis.” (L177-188-S)

16) Table S2: Consider making the font size on this table smaller (or decreasing the number of columns); numbers and words spanning multiple lines make this very difficult to read. (Same can be said about Table S4, though to a lesser extent.)

We have now used 9 font size and single line spacing in all tables in supplementary to make it easier to read. All supplementary tables now fit on a single page each.

Reviewer 2

The present study provides evidence that eggshell functionality evolves in relation to environmental pressures. The manuscript is very clear and it makes for a very interesting read. I only have some minor suggestions that I would like to see addressed before it is published.

We’re pleased the reviewer felt the manuscript was clear and interesting. We believe we’ve addressed all the minor issues, and thank the reviewer for the improvements to our paper.

Major comments:

1) It would be useful to know why (or why not) would one expect that in passerines the relationship between conductance and nest type, parental incubation be different to that previously reported by Portugal et al. 2014. In other words, is there anything fundamentally different between passerine egg or nest architecture/ incubation behaviour and to non-passerine reproductive traits?

Table 2 describes how different life-history traits may influence eggshell conductance. We predicted that passerines and near-passerines would have lower eggshell conductance than non-passerines, as passerines and near-passerines typically have altricial young and use exposed nests, while non-passerines have altricial and precocial young and are less likely to use exposed nests. The key differences between development mode and nest architecture between passerines, near-passerines and non-passerines are described in L197-206 of the Methods (pasted below), which could potentially be responsible for differences in eggshell conductance between these taxonomic groups.

“All passerines and near-passerines are land birds and have altricial and nidicolous (stay within the nest) chicks, while non-passerine chicks vary in their mode of development and include water and land birds [38]. Sandgrouse are an exception, as they have precocial young and are not tree-dwelling [39]. In respect to nest architecture, most passerines build open-cup nests, though some build more elaborate dome structures with roofs [40]. Dome nests, however, are more common among passerines than non-passerines, and are particularly frequent among very small passerines [41]. Although these groups are no longer recognised as near-passerines, this definition was used here to distinguish between ecologically profound differences among birds.”

Differences in embryo development and incubation period between altricial and precocial species, and how this may influence eggshell conductance in passerines and near-passerines versus non-passerines is described the last paragraph of the discussion (pasted below).

“Variation in incubation period across the altricial-precocial spectrum reflects a trade-off between embryo growth rate and degree of maturity when hatched. Precocial species take up to 2 times longer to incubate an egg of the same size as altricial species, but are far more developed when they hatch [75]. For eggs of the same mass, precocial species incur a higher total energy cost than altricial species because the embryo is larger for a longer period during incubation [76]. Consequently, eggs of species with fast (precocial) growing offspring had significantly higher RG_{H_2O} than those of species with slow (altricial) growing offspring based on top-ranked models. As higher conductance enables greater gas exchange, this may optimise embryo access to high energy content in precocial eggs [77], thus resulting in a more developed chick at birth. RG_{H_2O} in passerines was found here to be particularly low, likely because they have altricial young, whereas non-passerines consist of precocial and altricial species.” (L372-384)

2) It was not obvious until reading the results that the study spanned half of the avian orders and not only passerines (which I first thought, therefore the above comment). It would be useful to explain this since the introduction or methods.

Thank you for highlighting this. We have made additions to the Introduction to clarify this. We have added “spanning across 28 avian orders” to our study aims in the Introduction (L145) to increase clarity. We have also added a

paragraph in the Methods describing the key avian taxonomic groups explored in this study (L192-206), pasted below.

“Prior to updated avian phylogenies based on genomic DNA, near-passerines was a term given to tree-dwelling birds (within the conventional non-passerines) that were traditionally believed to be related to Passeriformes due to ecological similarities. In this study Pteroclitiformes (sandgrouse), Columbiformes (pigeons), Cuculiformes (cuckoos), Caprimulgiformes (nightjars), and Apodiformes (swifts, hummingbirds) were defined as near-passerines. All passerines and near-passerines are land birds and have altricial and nidicolous (stay within the nest) chicks, while non-passerine chicks vary in their mode of development and include water and land birds [38]. Sandgrouse are an exception, as they have precocial young and are not tree-dwelling [39]. In respect to nest architecture, most passerines build open-cup nests, though some build more elaborate dome structures with roofs [40]. Dome nests, however, are more common among passerines than non-passerines, and are particularly frequent among very small passerines [41]. Although these groups are no longer recognised as near-passerines, this definition was used here to distinguish between ecologically profound differences among birds.”

3) Although the sampling effort performed is commendable, sampled species account for ~4% of all bird species, this will always be a problem with comparative studies. How did the authors deal with the issue that results from having a large amount of missing data in the phylogeny, i.e. the outcome of the analyses is very sensitive to the inclusion of just a few extra species?

We agree that our study, as with most comparative studies, will have missing data in the phylogeny. Although we did not account for missing data directly in our analysis, we were extremely careful in selecting categorical predictors, ensuring a sufficient sample size for all factor levels.

Our top-ranked models identified multiple factors that influence eggshell conductance across a broad-taxonomic range of species, incorporating 28 avian orders. We acknowledge that incorporating a larger number of species (preferably by adding representatives from other avian orders, or additional species from extreme nesting environments to capture the full spectrum of life-histories) would improve the robustness of our statistical analysis, and help support our findings.

It is possible that the factors that were present but not significant ($p>0.05$) in our top-ranked models (e.g., eggshell maculation) may be important if eggs from other avian families or orders were available. However, we do not expect that the inclusion or exclusion of a few species from our current dataset would drastically change our results.

We would also expect the outcome to vary depending on the taxonomic scale. For example, comparative analyses on bird eggs across multiple orders (Stoddard *et al.* 2017; 2019) versus within families (Birkhead *et al.* 2019) identified different life-history traits contributing to the evolution of egg shape.

In view of this, we would encourage additional work on eggshell conductance variation within specific avian lineages, as broad-scaled patterns we observed are unlikely to apply equally to all smaller clades.

Birkhead, T. R., Thompson, J. E., Biggins, J. D., and Montgomerie, R. (2019). The evolution of egg shape in birds: selection during the incubation period. *Ibis* 161, 605–618. doi:10.1111/ibi.12658

Stoddard, M. C., Sheard, C., Akkaynak, D., Yong, E. H., Mahadevan, L., and Tobias, J. A. (2019). Evolution of avian egg shape: underlying mechanisms and the importance of taxonomic scale. *Ibis* 161, 922–925.

Stoddard, M. C., Yong, E. H., Akkaynak, D., Sheard, C., Tobias, J. A., and Mahadevan, L. (2017). Avian egg shape: form, function, and evolution. *Science* 356, 1249–1254. doi:10.1126/science.aaj1945

Minor comments

4) *I found the discussion slightly lengthy, perhaps some sections can be condensed. For example, last paragraph of page 13 (and first of page 14), also paragraph about calcium sources on page 14.*

Thank you for your advice on this matter. We agree that the discussion needs to be condense, so we have removed these sections from the discussion, as suggested.

5) *L130: change “to effect” with “to affect” (affect is the verb)*

This has been corrected. See L139.

6) *How many species were included in the measurements of conductance? (mention in the section “egg samples and preparation”.*

We have added at the start of Materials and Methods (L145): “In total, 365 bird species were included in this study.”

Figures

7) *Figure 1: I am a little skeptical about the usefulness of information presented here, perhaps is my personal unfamiliarity with this type of mapping, how was this performed? How reliable is it? Moreover, the continental differences are not further discussed or contrasted with the finding that temperature seasonality had an effect on conductance. However, seasonality would vary with latitude but not merely differ between North America and Europe I imagine? What can one take as a message from this information?*

This figure is provided to give a general indication of geographical variation in eggshell conductance based on the spatial distribution of species included in this study, and is not intended to be used for any analytical purposes. As this

distribution map is not paramount to this study, we have moved Figure 1 to Supplementary Information.

We used a modified R script provided on Dr Bruno Vilela's website (link: <https://rmacroecology.netlify.app/2018/01/23/a-guide-to-transform-species-shapefiles-into-a-presence-absence-matrix-based-on-a-user-defined-grid-system/> and <https://rmacroecology.netlify.app/2018/01/30/mapping-species-traits/>) to create the spatial distribution maps. Further information about the "letsR" package is available in their publication (Vilela and Villalobos 2015). We have provided the R script in Figshare (titled "R markdown_Species Distribution GH20 map.Rmd" in Spatial Distribution Traits folder), and we have described the process in section (i) Supplementary Information.

Vilela B and Villalobos F (2015). letsR: a new R package for data handling and analysis in macroecology. *Methods in Ecology and Evolution*. DOI: 10.1111/2041-210X.12401

8) Figure 2: This figure is impressive and aesthetically appealing but it is overloaded with information and difficult to interpret. The tree for example, displays the ancestral state estimation plus 5 variables. The magnitude of the bars from the two first variables is hard to perceive. Could this figure be split?

Following the reviewer's suggestion, we have split into two separate figures (now Figure 1 and Figure 2).

Figure 1 now focuses on variation in $\log(G_{H_2O})$ with body mass, which was the main significant predictor for this response variable in conditionally averaged models. We have increased the size of the circular bars plot around the phylogenetic tree so it is easier to interpret. Labels for each family are included in the phylogenetic tree.

Figure 2 shows the main significant predictors of residual G_{H_2O} . These were temperature seasonality and whether incubation was shared between parents. We have removed the family names from Figure 2a to reduce the amount of information. The scatter plot (Figure 2b) and hybrid box plot (Figure 2c) has been colour coded to match the ring around the phylogenetic tree (Figure 2a) identifying each categorical variable.